# Review of Flexible Wearable Sensor Devices for Biomedical Application

**DOI:** 10.3390/mi13091395

**Published:** 2022-08-26

**Authors:** Xueli Nan, Xin Wang, Tongtong Kang, Jiale Zhang, Lanxiao Dong, Jinfeng Dong, Peng Xia, Donglai Wei

**Affiliations:** 1School of Automation and Software Engineering, Shanxi University, Taiyuan 030006, China; 2School of Biomedical Engineering, Shanghai Jiao Tong University, Shanghai 200030, China; 3School of Mathematical Sciences, Shanxi University, Taiyuan 030006, China

**Keywords:** flexible wearable, biomedical, flexible materials, sensors

## Abstract

With the development of cross-fertilisation in various disciplines, flexible wearable sensing technologies have emerged, bringing together many disciplines, such as biomedicine, materials science, control science, and communication technology. Over the past few years, the development of multiple types of flexible wearable devices that are widely used for the detection of human physiological signals has proven that flexible wearable devices have strong biocompatibility and a great potential for further development. These include electronic skin patches, soft robots, bio-batteries, and personalised medical devices. In this review, we present an updated overview of emerging flexible wearable sensor devices for biomedical applications and a comprehensive summary of the research progress and potential of flexible sensors. First, we describe the selection and fabrication of flexible materials and their excellent electrochemical properties. We evaluate the mechanisms by which these sensor devices work, and then we categorise and compare the unique advantages of a variety of sensor devices from the perspective of in vitro and in vivo sensing, as well as some exciting applications in the human body. Finally, we summarise the opportunities and challenges in the field of flexible wearable devices.

## 1. Introduction

Flexible wearable devices, which target the flexibility of devices, have unique features and advantages, such as being light weight, having good flexibility [1,2], and potential for miniaturisation [3]. They are widely used in biomedicine [4], information acquisition [5], human–machine interaction [6], and robotics [7,8], triggering a new round of technological innovation in the biomedical device industry. The wearable devices that are applied to the human body aim to achieve the real-time monitoring of human body information, personalised diagnosis and treatment, and have great potential for further development. Convenient, at-home personalised medical devices can provide real-time information on human physiological conditions [9]; simultaneously, they can reduce the pressure on hospital treatment and can save resources. Wearable biomedical products are becoming increasingly popular and recognised by the public.

According to an industry survey report that was released by the Xinsijie Industrial Research Centre, the global biosensor market will reach USD 23.62 billion by 2021. It is expected that, from 2022 to 2026, the global biosensor market will increase at an average annual growth rate of approximately 9%. As the main downstream market, the medical field accounts for 78% of the demand and is primarily used in wearable medical devices. In the context of the new crown epidemic, various places have taken epidemic prevention measures, to a certain extent, which has stimulated the development of personalised wearable devices. Coupled with the aging population and an increasing prevalence of chronic diseases annually, biomedical wearable devices have ushered in huge development opportunities. Currently, the established wearable devices include smart watches, sports bracelets, hearing aids, and vision correction devices. These wearable devices play an important role in the real-time monitoring of physiological information, auxiliary treatment, and healthcare. We believe that, with the development of society, flexible wearable devices for biomedicine will become popular.

In the past few years, flexible sensing, which is the core of wearable devices, has enabled the quantification of conventional external stimuli [10], such as pressure [11], tensile forces [12], and temperature [13]. With the development of disciplines such as materials science, control science, and communication technology, there has been a series of breakthroughs in flexible electronics [14]. These sensors have achieved multi-angle and multi-directional information acquisition and can maintain excellent performance in complex environments [15], with varying degrees of sensitivity, minimum detection values, and detection ranges [16]. These performance enhancements provide better stability, speed, and accuracy for the detection of weak signals in the human body. Additionally, the development of implantable [17], adaptive [18], and degradable [19] materials and the emergence of self-powered bio-cells [20] have led to the emergence of a wide range of in vivo sensor devices. However, there is a lack of comprehensive summaries of flexible wearable devices in the context of in vitro- and in vivo-oriented biomedicine [21,22]; therefore, we present a systematic overview of a wide range of wearable devices from a sensor perspective, as well as some exciting applications.

The sensitivity of the sensor reflects the lowest detection limit of the device, corresponding to the signal strength that can be perceived and measured. The detection range corresponds to the sensing signal that can be applied, and the response time and relaxation time are related to the real-time and rapid response ability of the wearable device. These key metrics determine the applicability and sensing capabilities of the wearable device. Additionally, the durability [23], cycling stability, and biocompatibility [24] of the sensor are fundamental for the stable operation of wearable devices over time. These properties are widely regarded as essential characteristics of flexible sensor devices. The establishment of flexible wearable electronic systems with a high degree of flexibility [25] (stretching, bending, and folding) and excellent performance metrics (conductivity [26], sensitivity [27], detection range [28], and stability [29]) is necessary for the development of personalised medicine in society.

It is worth noting that flexible wearable sensors for biomedicine usually have excellent flexibility and compatibility, but also have low quality. The excellent flexibility can meet the requirements of a complex body surface, and the sensor deformation that is caused by different limb movements does not affect the performance of the device. Excellent compatibility can prevent adverse reactions (inflammation and allergy) between the device and human body; compatibility is a necessary property of implantable sensors. Low quality can be carried easily and increases people’s sense of experience, which is an important indicator of sensor production. However, wearable sensors for biomedicine still have limitations, and their production cost is relatively high, which is not conducive for the promotion of the products. There are many disposable wearable sensors that can cause environmental pollution. Cost reduction, recycling, and post-use degradation of sensors are important directions for current wearable-device research.

This paper has reviewed flexible sensor devices for biomedical applications, has summarised some synthetic materials that are commonly used for flexible sensing, has focused on their microstructure and sensing mechanisms, and has described the different ways in which sensor devices work in vitro and in vivo. The synthesis, doping, and modification of flexible materials exhibit excellent synergistic effects, enabling the integration of advantageous properties [30]. In vitro sensing is typically performed by using electronic skin [31] and medical devices [32] for human signals, whereas in vivo capture is typically performed using implantable and degradable [33] sensors for the information acquisition [34], wireless transmission systems [35,36], and self-powered electrical signal inputs [37]. Flexible electronics offer good compatibility with biological tissue materials, environmental adaptability, and harmless working behaviours to the human body, thereby promising a higher standard of disease monitoring and treatment. The general framework of this study is shown in Figure 1.

## 2. Selection and Production of Flexible Materials

The selection and fabrication of materials can significantly affect the performance of the sensors [38]. Biomedically oriented sensor devices place higher demands on the flexibility, conductivity, and durability of the materials. Existing flexible materials include polyethylene terephthalate (PET), polyimide (PI), polyvinyl alcohol (PVA), PDMS (Polydimethylsiloxane), and PEN (Polyethylene naphthalate).

### 2.1. Commonly Used Flexible Materials

PET, which is commonly used in flexible printed circuit boards, exhibits excellent mechanical strength. Simultaneously, PET has excellent electrical insulating properties and can be used as a device encapsulation layer. PET also exhibits light transmission properties and can be used as a protective layer for optical sensors.

PI has a few advantages over PET materials. Its biggest advantage is that it has excellent heat resistance. PI has excellent environmental stability and is unaffected by temperature or humidity. It can be used as a protective layer for sensors, preventing non-detectable stimuli from affecting the device performance.

PVA, which has excellent hydrophilicity, is often used in device adhesives and synthetic fibres. PVA has excellent film-forming properties, is easily degradable, and is popular in the manufacture of disposable devices.

Polydimethylsiloxane (PDMS) is the most commonly used flexible material. It has good flexibility and is an excellent force-sensitive material in both high- and low-temperature environments. Additionally, PDMS has good dielectric properties and can be used as a medium for pressure sensors. PDMS also has a certain air permeability, and is widely used in pressure sensing, flexible wearables, and device packaging.

PEN is commonly used in flexible printed circuit boards (PCBs) and capacitor films. It is more advantageous than PET in terms of heat resistance and mechanical strength and is a device material that comes in the form of a film. It exhibits good performance in blocking gas and water and can be used as a packaging material for devices.

Although PET, PI, and PEN are not flexible enough, these commonly used flexible device materials gain a certain degree of flexibility owing to the decrease in their thickness. PVA and PDMS are frequently used in patch-type wearable sensors.

However, these materials cannot meet the requirements of sensor fabrication. The synthesis, doping, and modification of a variety of flexible materials that exhibit excellent synergistic effects (1 + 1 > 2), and the integration of advantageous properties resulting from the reaction, are major current studies in the field of material synthesis [39]. This paper summarises the selection and the fabrication of the following three materials: carbon nanomaterials [40], polymeric materials [41], and hydrogel materials [42]. The typical examples of material syntheses are listed in Table 1.

### 2.2. Carbon Nanomaterials

Carbon nanomaterials, which exhibit excellent electrical conductivity, such as graphene [50], carbon nanotubes [51], and MXene [52], have received considerable attention from scientists because of their excellent performance indicators and unique nanostructures. In more than 10 years since the discovery of graphene, the organic combination of different biomolecules and graphene has shown excellent functionalisation and a strong carrying ability, which stems from its ultra-high lateral space and excellent electrical conductivity. Carbon nanotubes and MXene have a natural advantage in the hybridisation of carbon-based materials because of their considerable biocompatibility, their inherent degradability of carbon-based materials, and their matching electrical conductivity. The difference is that MXene is more likely to produce different biological effects, owing to its hydrophilicity, diffusivity, and controllability caused by functional groups [53], which is particularly important in biomedicine. The excellent stability of carbon nanomaterials [54] makes them uniquely competitive in device cycle testing.

Lee et al. evaluated a new dopant-induced heterodimensional hybridisation method for 1D/2D materials (Figure 2a) by constructing hybrid structures of 2D Ti3C2Tx-type MXene and 1D graphene nanoribbons through nitrogen doping [43]. The structure has an excellent viscosity, a high resistance, and fabricates as a piezoresistive pressure sensor with hysteresis as low as 1.33% and a stable cycle count of more than 10,000 cycles under high pressure. Particularly, the authors have improved the connection properties of a low-dimensional hybrid material [55] via elemental doping in a gaseous environment, which resulted in better interfacial adhesion and substantially improved the hysteresis performance and high-pressure cycling stability of the sensor.

Ding et al. reported a self-assembled 2D carbon nanostructure network (Figure 2b) that was based on an electrospray/mesh technique [44]. Low-dimensional carbon nanomaterials exhibit excellent optoelectronic properties; however, their nano properties are significantly degraded when they are assembled into blocks. By controlling the dynamic injection of charged droplets and combining the nanoscale diameter of 1D carbon nanotubes and the lateral infinity of 2D graphene, a 2D nanostructured network was formed. It can be used as a pseudo-3D structure that displays an ordered network of nanofibres [56] and is laterally infinite, with excellent flexibility and nanoscale electrical conductivity.

Yang et al. prepared a Ti3C2Tx-type MXene film that was resistant to oxidation (Figure 2c) in a liquid environment [45], and formed a stable hydrophobic protective interface via fluorine functionalisation. Thus, by embedding bacterial nanocellulose, a highly sensitive sensing medium layer can be prepared with excellent detection performance in pressure sensing. In a liquid environment, its conductivity remains almost unchanged, and this excellent waterproofing performance is expected to realise invasive-force detection. The excellent hybridisation ability [57] and unique sensing properties [58] of MXene have attracted the attention of scientists in the field of biosensing.

These carbon nanomaterials have been successfully hybridised using a unique synthetic approach in order to obtain low-dimensional carbon materials, successfully expanding the 2D structure of carbon nanomaterials and enriching the potential mechanisms of the structure–property relationships [59]. It is worth noting that, when carbon nanomaterials are used alone, especially in bulk, their nanometre properties are severely compromised. Carbon nanotubes are prone to an uneven distribution during their application. Ti3C2Tx-type MXenes also have the problems of easy oxidation and recombination, and the hybrid interface is unstable. These limitations require the reasonable doping of carbon nanomaterials in order to optimise the device performance. However, the excellent properties of carbon nanomaterials, including adsorption, electrical conductivity, thermal conductivity, flexibility, low mass, and a high specific surface area, make them irreplaceable in biomedical wearable sensors.

### 2.3. Polymeric Materials

Polymeric materials [61] are often used as flexible substrates and coating materials in order to improve the performance of sensors during fabrication. Polymer materials have been doped and modified in order to achieve different types of organisation and unique properties, including higher electrical conductivity [62] and ion mobility [63], as well as more stable structural and mechanical properties.

Recently, Ejima et al. reported a non-standard phenolic polymer [46] that exhibited strong underwater adsorption (Figure 2d). The addition of a hydroxyl group to the catechol moiety enhanced the adhesion capacity. The synthesis of phenolic polymers with four or five hydroxyl groups on styrene monomers results in superb underwater adsorption. These non-standard phenolic groups have excellent adsorption properties and need only be added to polystyrene in small quantities to produce strong underwater adhesion on different substrates. Particularly, this adhesion exhibits excellent stability and durability for long-term applications in bonding devices in liquid environments.

Tao et al. reported an ultrafine polyaniline fibre [47] that was wet-woven using a solvent-exchange strategy (Figure 2e). The polymer fibre reduced the viscosity of the fibre by diffusion between the solvents while enhancing the tensile capacity of the fibre. During stretching, the polymeric fibre maintained excellent mechanical properties, good electrical conductivity, and was less than 5 µm in diameter. The fibre is an excellent electrode material, which is even better than some carbon nanomaterials in energy storage [64] and current conduction [65]. The stability of its electrochemical performance is expected to be a major breakthrough in the self-supply of the system.

These polymeric materials have been developed in order to provide a more comprehensive range of performance requirements and to have significant applications in sensor integration and wearable fabrics [66]. Biomedical polymer materials are rapidly developing in terms of functionalisation, refinement, and fibrosis. The synthesised polymers have achieved remarkable results in implantable, degradable, catalysed, energy storage, and energy conversion applications, and are widely used. However, it is difficult to change the complexity of the preparation of polymer materials and the high cost; the effective improvement of this limitation will bring a powerful boost to the promotion of the research products.

### 2.4. Hydrogel Materials

Small and thin flexible devices should be more compatible and intelligent as alternatives to traditional medical devices. Carbon nanomaterials and polymeric materials do not fully meet the requirements of future personalised medical devices. The emergence of modified hydrogels [67] provides more options for personalised medical devices, which are widely used in areas such as electronic skin and human–machine [68] interactions. The porous structure of hydrogels is conducive for enhancing the transfer rate at the interface and has a wide range of applications in ion transfer and signal transmission [69]. The prepared hydrogels are extremely biocompatible by dissolution, reaction, and modification in order to obtain various advantageous properties [70].

For example, Kim et al. developed an ultra-soft, highly permeable, low-impedance ultra-thin hydrogel (Figure 2f) that acts as a liquid electrolyte on the skin, forming a tissue-like spatial- and quasi-solid interface [48]. This interface has a high permeability and low impedance properties and is extremely ’conformable’ to different body parts. The porous structure [71] realises the diffusion and propagation of biological macromolecules, and the low impedance improves the injection efficiency of charges under external stimulation. Additionally, Wang et al. proposed a method to adhere hydrogels to various solid interfaces [60], providing a conformal contact method between the hydrogel and the torso sites (Figure 2g). The hydrogel was dehydrated, was glued on to a solid interface with a small amount of glue and was hydrated in order to form a new gel layer. This method enables the organic bonding of hydrogels to human tissues and has great potential for applications in bioassays and healthcare. Moreover, the conformal adhesion [72] of hydrogel significantly changes the physicochemical properties and electrical conductivity of the solid surface, making it possible for the same device to be reused in different complex environments.

Qiu et al. developed a highly soluble, hyperbranched nanoparticle-reinforced polymer hydrogel (Figure 2h) with no temporary entanglement, hysteresis-free material properties during cyclic loading, and fatigue-free properties that can significantly reduce energy dissipation and can exhibit excellent stretchability and flexibility [49]. Minimal material energy dissipation [73] is essential for achieving high device performance at sites of repetitive human actions (e.g., heartbeat, breathing, and movement). In hydrogel networks, hyperbranched nanoparticles act as the dominant cross-linked, highly expanded polymer chains. They are linked without temporary entanglement [74] and have excellent elastic properties, exhibiting great potential in strain and impedance sensors.

As special polymer materials, hydrogels exhibit extremely competitive properties in terms of biocompatibility. Hydrophilic polymers are often used as functional groups in hydrogels, which produce different physical and chemical properties under different environmental stimuli through physical or chemical cross-linking modification. The swellability of the hydrogel medium and the resulting network voids are greatly beneficial for the penetration and transportation of macromolecules. Additionally, hydrogels inherit the disadvantage of the high cost of polymer materials, while increasing the complexity of integration. It is undeniable that, although hydrogel is slightly insufficient in terms of cost and yield, its performance is excellent and cost-effective.

In summary, carbon nanomaterials, polymeric materials, and hydrogel materials were doped and synthesised rationally in order to obtain exciting and excellent properties. The difference is that carbon-based materials tend to acquire excellent piezoelectric properties [75], whereas hydrogel materials tend to focus more on the flexibility and permeability [76] of the material. The recently emerging metal-organic frameworks (MOFs) [77] are also highly sought after, and possess functionalities that are similar to carbon nanomaterials and porous structures that are similar to hydrogel materials. The different organic frameworks can be formed by self-assembly. Cu-based [78], Zr-based [79], porphyrin-based [80], and MXene-derived [81] MOFs have emerged and have begun to show their cutting-edge applications in biosensing, medical detection, and gas monitoring. The development of MOF materials with different groups is expected to achieve further breakthroughs and optimisation in the field of biomedical device fabrication. It is worth noting that different materials may produce nonlinear deviations under different working environments, and the choice of a reasonable working environment for synthesis is significant for the retention of different advantageous properties and the modification of inferior ones, which may result in unexpected outcomes.

## 3. Application of Sensors in the Human Body

Flexible wearable sensor devices are widely used in human–machine interactions, healthcare [82], and personalised diagnosis and treatment systems [83] in order to determine the physiological level of the human body. The prevalence of various high-precision preparation processes, such as 3D printing [84], photolithography [85], and printed coating techniques [86], has dramatically increased both the accuracy and the yield, facilitating the ease of device fabrication. The use of other auxiliary equipment and test systems, such as centrifuges, thermostatic heating chambers, ultrasonic devices, and LCR testers, enables more efficient device fabrication. Simultaneously, wearable devices should have good flexibility, transportability, and excellent biocompatibility [24], contributing to more sensitive feedback, stable control, and efficient transmission. From the perspective of the detection location, sensors can be classified as in vitro [87] and in vivo [88]. In vitro sensor devices are further divided into electronic skin classes and other medical devices based on the appearance of the package and the application of the device. Compared with in vitro sensing, the application environment [89] for in vivo sensing is more stringent, but many professors and scholars have made unique attempts and products. In this review, we present a comprehensive overview of in vitro and in vivo sensing techniques.

### 3.1. In Vitro Sensing

In recent years, wireless communication [90] and human–machine interaction technology have developed rapidly, and signal transmission is highly consistent and instantaneous, making the in vitro sensing of wearable devices possible. In vitro flexible wearable devices that are used to track the movement of various parts [91] of the human body, assist in the diagnosis of various diseases [92], and can establish a higher level of human–machine interaction interface, have become a major focus in in vitro sensing. Wireless medical devices that are based on wireless data transmission, which can be applied to flexible platforms and environments with excellent information output and control stability, have become one of the main trends in biomedical fields. In vitro sensor devices usually exist in the form of electronic skin or medical devices. As such, flexible wearable devices can adapt to human limb movements and have good bending and stretching properties [93].

#### 3.1.1. Relying on Electronic Skin for In Vitro Sensing

Electronic skin serves as an excellent integrated platform with a thin and soft interface that can be applied to most parts of the human surface. Electronic skin can sense internal physiological information about the human body, can make sound judgments, and can prevent and monitor diseases. The inherently weak properties of the human signal dictate that the built-in sensors must be sufficiently sensitive [94], and the persistence of the human signal requires an excellent robustness [95] of the electronic skin. The sensor transmits the human body signal as an optoelectronic signal in order to enable information interactions.

Deng et al. reported a patch-type piezoresistive sensor with both high sensitivity and a wide detection range (Figure 3a) for monitoring human physiological signals [96]. Inspired by the microstructure of a rose petal, the sensor exhibits excellent mechanical flexibility and electrical properties using a layered microstructure with polyaniline/polyvinylidene fluoride nanofibre films on the top and bottom layers and interlocking electrodes with a domed structure in the middle. Structurally, this domed interlocking structure [97] can significantly increase the contact area and reduce the resistance when subjected to pressure and exhibits a low relaxation time, enabling a sensitive adjustment of the electrical signal output of the sensor (Figure 3b). The structure can significantly expand the contact area of the nanofibres under pressure and can detect small pressures. In terms of performance, the sensor has a sensitivity of 53 kPa-1, a response time of 38 ms, a recovery time of 19 ms, a very low power consumption, and an excellent robustness (over 50,000 cycles). The designed skin patch has a significant ability to perceive vibrational forces and can monitor subtle motion signals (e.g., breathing, swallowing, muscle vibration, and foot pressure).

Wen et al. designed an electronic skin that senses pressure and temperature independently (Figure 3c) and can convert pressure into a voltage difference between polar plates [98]. Similarly, the sensor uses a tapered microstructure for sensitive pressure sensing. In contrast, the sensor employs a triboelectric nanogenerator (TENG) whose tapered structure is applied to the triboelectric layer. Particularly, the electronic skin can express the temperature signal independently while also sensing pressure (Figure 3d). The temperature signal is detected using a specially designed thermocouple membrane, and the thermoelectric coupling effect [99] can be expressed. The triboelectric sensor senses the pressure change from the vertical direction, whereas the thermocouple film senses the temperature change from the horizontal direction. This vertical detection method avoids the complicated decoupling process, enabling dynamic sensing of pressure and temperature from external signals.

In the above-mentioned electronic skin research, the perception of the magnitude of the force was realised, but the perception of the direction of the force was not specific. This is because the relatively parallel hierarchical structure of the electronic skin has unique advantages in terms of the perception of force magnitude, and the perception of force direction [100] often needs to be performed through an array. Therefore, integrating sensor units into electronic skin arrays is a widely used approach in sensor research.

**Figure 3 micromachines-13-01395-f003:**
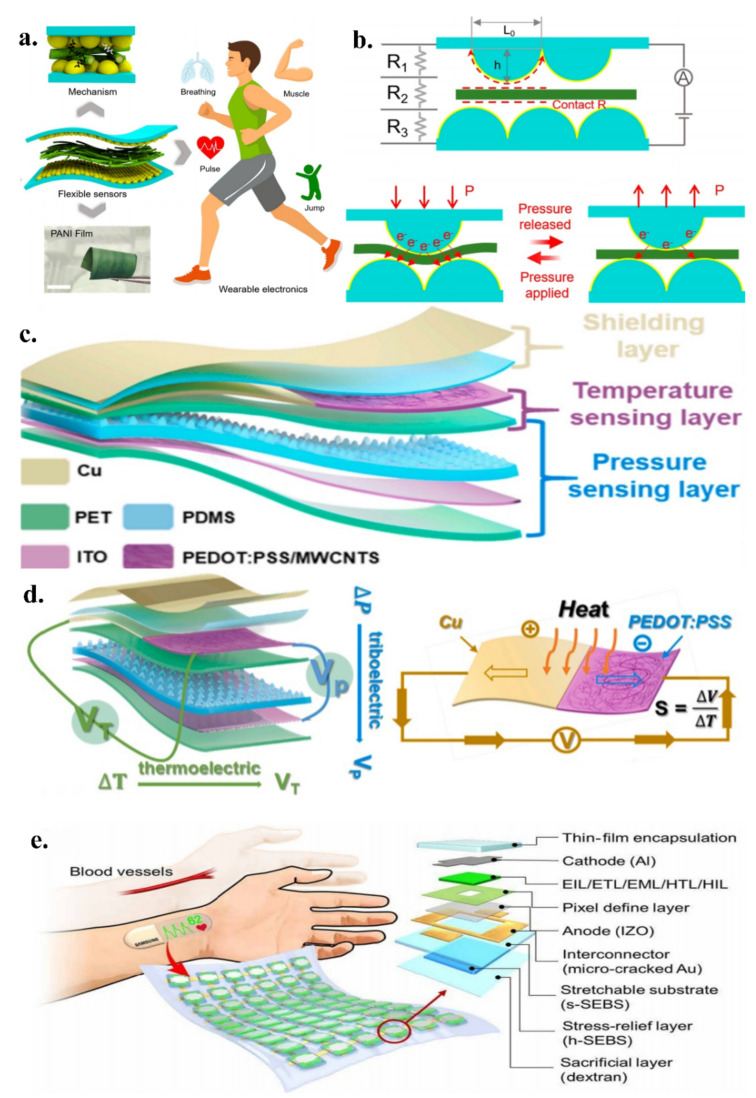
(**a**) Patch−type piezoresistive sensor; reproduced with permission from the American Chemical Society (2021) [96]; (**b**) Electrical signal output from the sensor; reproduced with permission from the American Chemical Society (2021) [96]; (**c**) Electronic skin that senses pressure and temperature independently; reproduced with permission from Elsevier (2022) [98]; (**d**) Temperature signal from the electronic skin; reproduced with permission from Elsevier (2022) [98]; (**e**) A self-contained skin−like health care patch; reproduced from Ref. [101].

Notably, Yun et al. reported a stand-alone skin-like health care patch (Figure 3e) that detects changes in the heart rate through conformal contact with the wrist [101]. The body of the patch is a flexible light-volume tracing sensor array consisting of an organic photodiode (OPD) and two organic light-emitting diodes (OLED). The OLED emits a red light. After being reflected by the arm, a change in the reflected light intensity is detected by the OPD. Notably, the intensity of the reflected light varies owing to the changes in the heart rate. It also carries a soft display consisting [102] of a diode array (17 × 7) with a variable microcrack. Au interconnects combined with a stress relief layer in order to allow stable operation under folding, twisting, and 30% stretching. The signal is processed by the microcontroller, is filtered, counted, and displayed on the diode array in real-time. The patch can be scaled up to monitor biosignals in daily life, contributing to more comprehensive health monitoring.

These studies on sensing human signals have demonstrated that sensors rely on electronic skin in order to establish a sensing platform [103] for human signals, which can effectively capture human body information and is extremely important for implementing the health monitoring of the human body. However, this information does not enable a comprehensive judgment of human health, and the monitoring of some hidden signals, particularly in vivo signals [104], is one of the current research areas in in vitro diagnostics.

In addition to these traditional patch-type electronic skins, the application of electronic skin with integrated triboelectric nanogenerators (TENGs) has recently emerged, which realises the self-powering of wearable devices. The application of TENGs makes the integration of wearable devices and other sensor communication technologies more convenient, and the self-powered feature has unique advantages for device miniaturisation.

Recently, Wang et al. reported an intelligent tactile sensing system combining TENGs and deep learning technology [105] (Figure 4a), which has an important practical value in the cognitive learning of visually impaired patients and the fabrication of bionic prostheses. The system integrates three separate tactile sensors that are doped with different materials and uses a raised surface structure for better sensitivity and perception. By extracting features from the electrical signal of the tactile sensor and through coupling calculations, the typical features of an unknown material can be obtained. The relationship and normalisation of the three electrical signals were established in order to realise the tactile perception of size, position, pressure, temperature, and humidity. By integrating a convolutional neural network, the features were visualised, and a high-precision detection of 96.62% was achieved. The application of deep learning technology can achieve more subtle material feature recognition than that of the human eye.

Wang et al. designed a badge-reel-like, stretchable, wearable, self-driving sensor, and its system [106] (Figure 4b). TENGs, utilising grating structures, exhibit extremely low hysteresis and ultra-high durability (over 120,000 duty cycles). When the sensor is bent or stretched, the triboelectric layer is squeezed in order to produce an electrical signal. On this basis, the use of peak counts for the dynamic monitoring of body posture, especially in places where bending and stretching of the spine, joints, neck, and other locations often occur, is conducive for reducing the risk of disease in the spine and joints that may be caused by long-term bad posture. The system adopts mature 3D printing technology and flexible printing technology, making it possible to realise the popularisation of the products.

These studies on the perception of external stimuli have proven that flexible electronic skin can effectively interact with the external environment. On the basis of breaking the original balance by external stimulation, the triboelectric layer is contacted and squeezed in order to realise the output of the electrical signals. The TENGs enable the e-skin to achieve self-sufficient power requirements, which can quickly provide detectable responses. Additionally, avoiding the influence of interfering stimuli is still an urgent problem that needs to be solved, which determines the environment in which they can work.

#### 3.1.2. Relying on Medical Devices for In Vitro Sensing

Flexible medical devices [107] are worn on the human body and they interact with information using built-in sensors. Compared with electronic skins, medical devices have a variety of packaging forms and can carry sensors with various geometrical appearances [108], more diverse sensing forms, and more comprehensive sensing information, promising better diagnosis and treatment.

Wang et al. designed an active sensing array [109] for the diagnostic assessment of lumbar degenerative diseases by detecting plantar pressure distribution (Figure 5a). A PVDF piezoelectric sensor was used to convert force to electricity. The array performed well in pressure cycling and response time tests with a high degree of robustness, supported vector machine supervised learning algorithms [110], collected walking foot pressure distributions from patients with lumbar degenerative diseases for common human limb movements, built a database sample, classified and identified based on plantar pressure, and achieved a recognition accuracy of 99.2%. The monitoring system that was established based on the sensing array allows information interaction with mobile phones via Bluetooth in order to display the wearer’s foot pressure distribution in real-time. The patient’s foot pressure data can be effectively uploaded to the client. It facilitates the diagnosis and the rehabilitation assessment of patients with degenerative lumbar spine diseases. This back-end data processing method has been explored for the establishment of the medical Internet of Things.

In another study, Koh et al. reported a flexible gripper [111] that is capable of gripping millimetre-sized organisms while enabling temperature and pressure sensing (Figure 5b). The gripper enables bidirectional sensing and stimulus interaction through silver nanowires [112], while acting as a mediator for Joule heating. The strain and the vibration are detected through the addition of a cracked strain sensor, which changes the structure of the strain gauge by stretching and bending the gripper. This crack-type strain sensor is located on the back of the gripper, and any mechanical deformation that is experienced by the soft gripper can be sensed, which causes structural changes in the crack and affects the resistance of the sensor. By carrying a heater, temperature sensing and thermal motion are realised using AgNWs. Additionally, the gripper can provide temperature and pressure stimulation, which can contribute to the maintenance of human tissues, such as muscle soreness and bone misalignment.

**Figure 5 micromachines-13-01395-f005:**
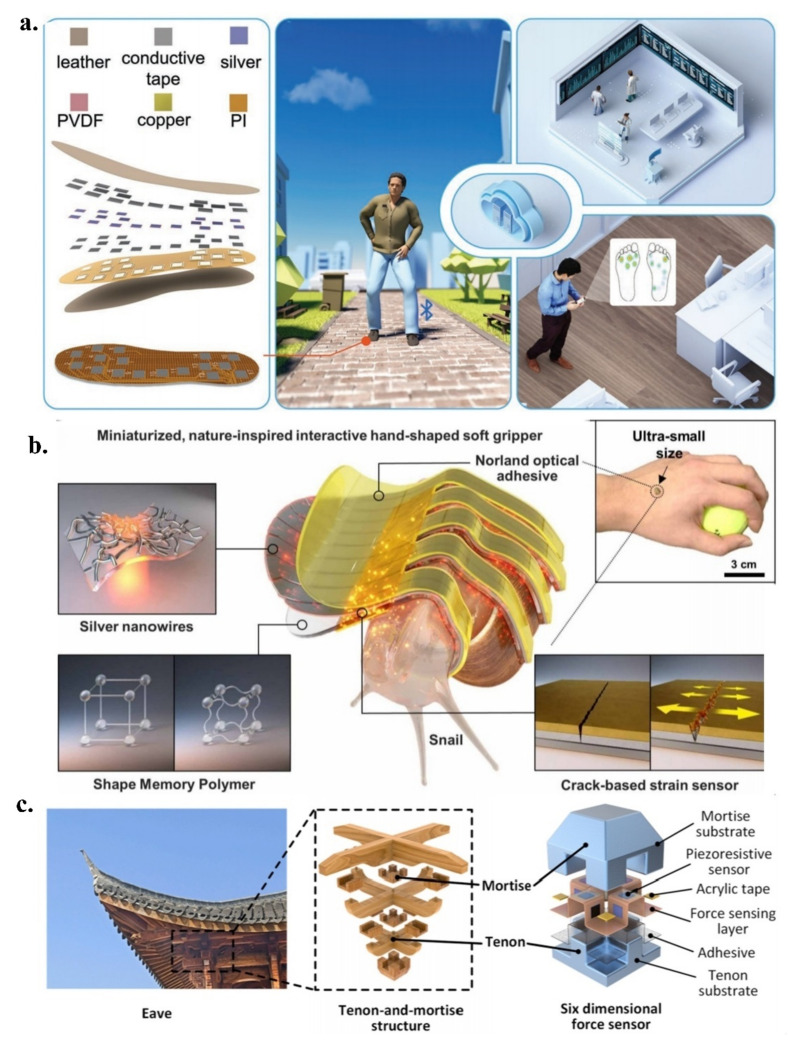
(**a**) Active sensing array; reproduced with permission from Wiley (2022) [109]; (**b**) Flexible gripper; reproduced with permission from the American Association for the Advancement of Science (2021) [111]; (**c**) Six-dimensional force sensor with a flexible mortise and tenon interlock structure; reproduced with permission from Elsevier (2022) [113].

Wu et al. reported a six-dimensional force sensor with a flexible mortise and tenon interlocking structure (Figure 5c), which was studied and decoded using a neural network [113]. The six-dimensional forces are forces F_x_, F_y_, and F_z_ and moments M_x_, M_y_, and M_z_. From the perspective of control science, the sensor performs neural network and finite element analyses on the forces at different angles, showing that the sensor has different states under multi-angle forces. The sensor, with a minimum size of 7 mm × 7 mm × 7 mm, is used in the precise orthodontic treatment of teeth. It has 12 sensing units that can quantitatively detect external stimuli in different directions and output a voltage signal when the sensor is squeezed or twisted.

In addition to these traditional force tests, “non-force” tests are favoured by researchers. For example, temperature detection [114] of the human body surface and body and the detection of breathing gas [115]. In this era of the COVID-19 pandemic, these tests have attracted special attention in medical surveillance.

Recently, Ding et al. reported a fibre-optic temperature sensor [116] that is based on photoelectric up conversion (Figure 6a), which exhibits strong temperature dependence from the conversion of infrared light to visible light. The sensing unit is mainly composed of InGaP-based double-junction light emitting diodes and GaAs-based light emitting diodes. A large bandgap is connected in a series in the middle, exhibiting excellent sensitivity to temperature. With an increase in the temperature, the optical wavelength also increases. Simultaneously, an increase in the temperature leads to the narrowing of the bandgap, which leads to a reduction in the voltage. The optical signal exhibited excellent robustness in the cyclic heating and cooling tests. Additionally, the sensor has a strong anti-electromagnetic interference ability, and when combined with thermal imaging [117] and nuclear magnetic resonance [118], it is expected to be able to measure the local temperature in the body. The wavelength of the emitted light of the sensor is still limited to the red-light band, and the development of emitted light in the longer wavelength band is a research hotspot in the field of photoelectric up conversion in the future.

Sheng et al. reported a polyimide-sputtered and polymerised [119] gas humidity sensor (Figure 6b); it broke the perception that the traditional polyimide capacitive humidity sensor has low sensitivity and a slow response, and at a large detection range, low humidity (<40%) could still achieve a humidity measurement. Polyimide-sputtered and polymerised materials have excellent hygroscopicity and hydrophilicity, and their porous spatial structure realises rapid gas transfer. When the sensor is in a humid environment, the film is gradually oxidised, which directly affects the capacitance of the sensor, thereby realising humidity detection. The sensor is applied to respiratory humidity monitoring [120], which detects the humidity of inhaled and exhaled air, and is very competitive in the mask, face shield, and helmet industries.

**Figure 6 micromachines-13-01395-f006:**
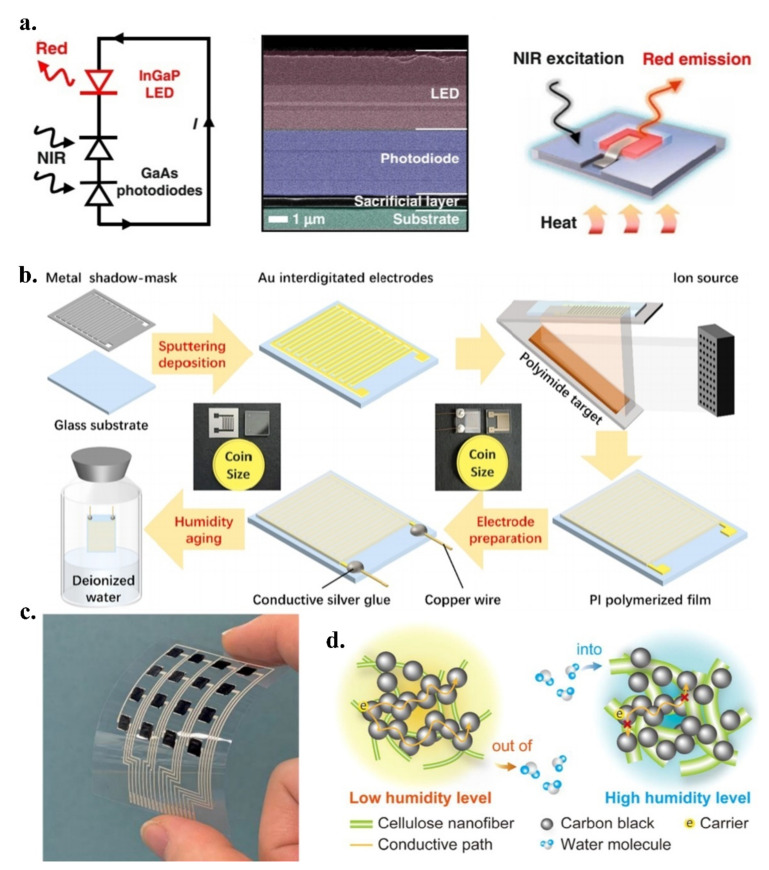
(**a**) Fibre-optic temperature sensor based on photoelectric up conversion; reproduced with permission from Springer Nature (2022) [116]; (**b**) Polyimide-sputtered and polymerised gas humidity sensor; reproduced with permission from the American Chemical Society (2022) [119]; (**c**) Ink-based printing flexible humidity sensor; reproduced with permission from the American Chemical Society (2022) [121]; (**d**) Schematic of the humidity detection mechanism for the cellulose nanofibres/carbon black composite; reproduced with permission from the American Chemical Society (2022) [121].

Similarly, Tokito et al. reported an ink-based printing, low-cost, fast-fabricated, and flexible humidity sensor [121] (Figure 6c). Conductive ink, which is composed of cellulose nanofibres and carbon black, changes the resistance of the ink according to the change in humidity (Figure 6d). The hygroscopic expansion of nanofibres destroys the conductive path of the carbon black, resulting in an increase in electrical resistance. The porous physical structure of nanofibres makes this hygroscopic ability even better. Additionally, the carbon black content is critical for the conductivity of the ink. Low content prevents the ink from forming sufficient conductive paths, whereas high content reduces the sensitivity of the sensor. In a high humidity environment, the ionic conductivity of the ink is expected to achieve sensitive humidity detection. The sensor realises low-cost and high-efficiency humidity detection and has unique advantages in the disposable product manufacturing industry. It is worth noting that the adhesion between the sensing layer and substrate is insufficient, and it can easily fall off in a humid environment. The combination of this with the non-standard phenolic polymers that have been mentioned in the previous materials section, perhaps with unexpected effects, would be interesting.

Sensors play a significant role in the research on medical devices. The combined use of multiple sensors allows the detection of information regarding the human body from multiple perspectives and provides a more comprehensive understanding of human physiology. The application to different parts of the human body [122], particularly to uneven areas, in a flexible package [123] creates conformal contact with the body. The development of these electronic skins and the design of medical devices have significantly improved the diagnosis and treatment methods of human diseases, opening a ‘new window’ in the future of smart medicine.

### 3.2. In Vivo Sensing

The inherent mechanical and biological properties of human tissues vary among people, and implantable devices that achieve immediate detection face major problems [124], such as the following: (1) it is difficult to apply existing electronic devices to organs with 3D structures, which can easily cause rejection by the body; (2) the devices are in a complex environment in the body, and long-term monitoring may lead to device failure; and (3) there are difficulties in achieving a stable transmission using passive wireless reading devices for strain sensors and maintaining the quality factor of the RLC circuit. Additionally, the complex physiological environment inside the human body, the secretion of stomach acid and catabolic enzymes, and the inherent contraction and expansion movements of organs and tissues can cause damage to the device structure and the human body. In vivo sensor devices are often implantable, adaptive, and degradable, allowing for in vivo ‘spaced’ sensing [125,126], wireless charging [127], and self-powered power devices [128,129,130], which are harmless and adaptable in vivo. The device is stable in the event of an in vivo organ ‘movement’ and can be displayed in real-time on a mobile device terminal via Bluetooth [131].

Recently, Lee et al. reported a suture-process-connected, wireless capacitive fibre optic strain sensor (Figure 7a) for connective tissue monitoring [124]. The sensing system consists of a hollow double-helix structure comprising two stretchable conductive fibres combined with a passive RLC circuit. The relationship between the mechanical and electrical properties is derived using an analytical expression. The hollow double-helix structure forms excellent tensile properties. During the stretching process, the two fibres approach each other and are squeezed after contact, resulting in a change in the capacitance. The addition of an induction coil to the wireless system realises the wireless signal-reading capability. The mechanical–electrical property relationship can be adjusted through the conductive fibre length diameter, creating more relaxed conditions for the implantation position, and the system enables wireless transmission without welding [132].

Jeong et al. proposed a wirelessly charged, fully implantable soft optoelectronic system [133] (Figure 7b), which realises rapid adjustment and adaptation in complex environments and is widely used in in vivo neuroscience research. The system performs a light stimulation through optoelectronic nerve probes, realises wireless transmission through coil antenna inductive coupling, receives radio-frequency energy, and achieves wireless charging. The system simultaneously enables remote selective wireless control and closed-loop wireless automatic charging capabilities for multiple animals using an off-the-shelf smartphone. In the preparation of devices, the development of phenolic polymers, which have a strong adsorption force in liquid environment, has made outstanding contributions to the bonding and integration of devices. Phenolic polymers are often used as carriers for drug delivery and have excellent antibacterial and anti-inflammatory properties [134]. This enables long-term in vivo studies without intervention [135], creating a new benchmark for in vivo sensing. However, the system suffers from metal-sensitive interconnects and uses batteries, resulting in a certain degree of incompatibility with MRI machines, narrowing its potential for biomedical applications; however, the design is still good enough.

**Figure 7 micromachines-13-01395-f007:**
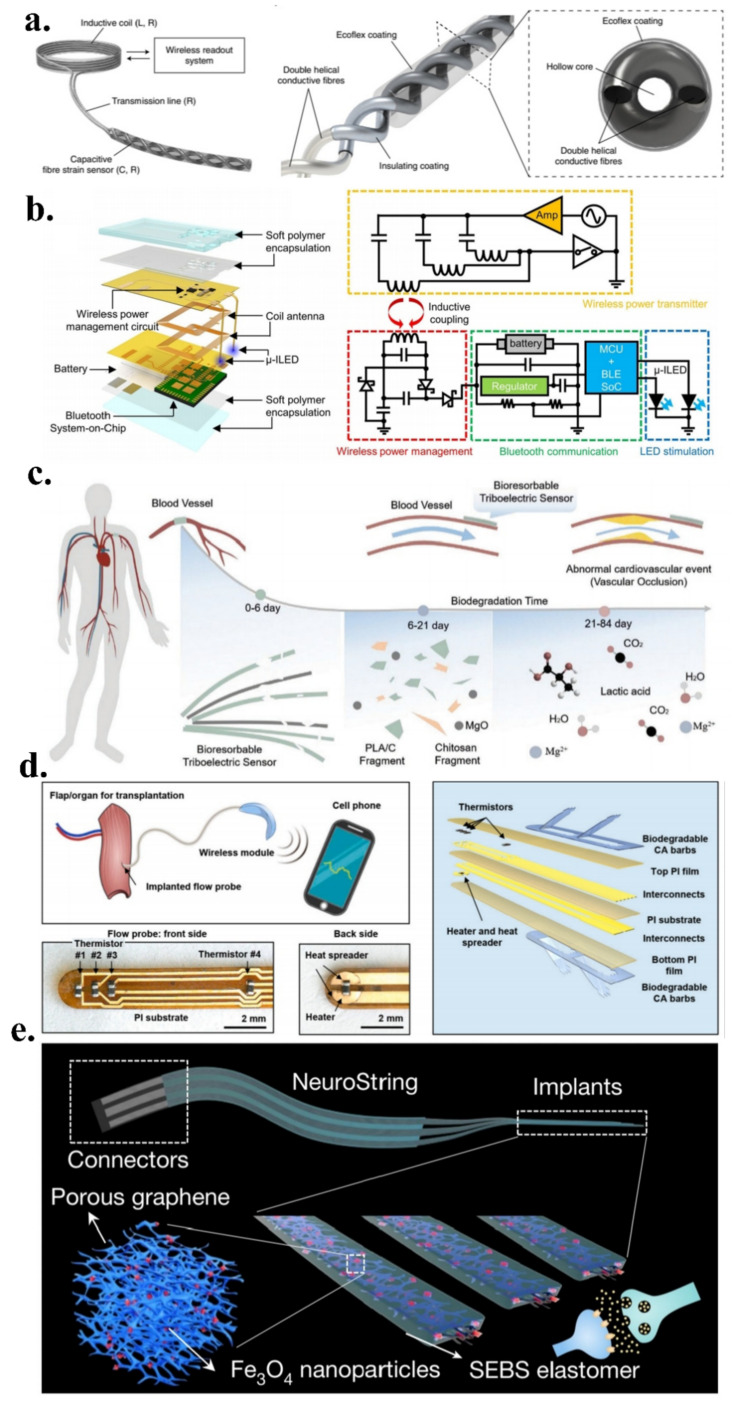
(**a**) Suture–process–connected wireless capacitive fibre optic strain sensor; reproduced with permission from Springer Nature (2021) [124]; (**b**) Wirelessly rechargeable, fully implantable soft optoelectronic system; reproduced with permission from Springer Nature (2021) [133]; (**c**) Biodegradable self–powered pressure sensor; reproduced with permission from Wiley (2021) [136]; (**d**) Wirelessly implantable microflow sensing system; reproduced with permission from Elsevier (2022) [137]; (**e**) Stretchable biological interface for neural tissue; reproduced with permission from Springer Nature (2022) [138].

In another study, Wang et al. reported a biodegradable, self-fed pressure sensor (Figure 7c) that could convert environmental pressure into an electrical signal for postoperative cardiovascular care [136]. The core of the sensor consists of two triboelectric layers, which are in contact under the action of an external force in order to achieve voltage output. An air space structure was introduced in order to ensure that the force–electricity change curve had good linearity. The sensor exhibits good sensitivity and durability by providing kinetic energy through the frictional electrical effect of the degradable material. Additionally, the sensor had 99% antimicrobial properties, a 5-day service time, and an 84-day degradation time, which can effectively prevent wound infections.

Rogers et al. reported a wireless implantable microflow sensing system (Figure 7d) that enables stable and reliable flow sensing with readout on a mobile terminal [137]. The sensor detects the blood flow through a heater and thermistor and is connected through a small Bluetooth module that is attached to the skin for data reading and for visual display on the mobile terminal. The sensing system was applied to surgical flaps and later, to organ transplantation in order to detect bleeding in microvessels using a thermal probe for continuous monitoring. The probe barbs exhibit excellent resistance to temperature and mechanical disturbances, are made of degradable materials, facilitate the mitigation of inflammation in organ tissues, and exhibit strong biocompatibility.

Interestingly, Bao et al. evaluated a stretchable biological interface [138] for neural tissue (Figure 7e), which was prepared by the laser patterning of a metal-complexed polyimide into an interconnected graphene/nanoparticle network that was embedded in an elastomer. The interface is used for a seamless connection between the central nervous system and gastrointestinal tract tissue. The real-time detection of neurotransmitters is realised through different impedances in different stretching states, and autonomous monitoring can be performed without external stimulation. The interface has excellent stretchability and softness and is highly compliant with intestinal tissue, avoiding the unnecessary irritation that is caused by peristalsis. Moreover, the interface has excellent adhesion and easily adheres to tissue membranes. It can also be mounted on endoscopes in order to realise direct sensing of the stomach, which is expected to solve the problem of the correlation between microorganisms and intestinal chemical kinetics [139].

These studies of implantable sensing systems imply that in vivo sensing has unique advantages in biomedicine, enabling more direct information acquisition [140] and health monitoring [141]. Additionally, implantable sensor devices are uniquely positioned to carry drugs [142,143] and can provide promising targeted drug delivery for localised treatment. This near-attrition-free drug therapy significantly enhances the treatment efficiency and is a novel clinical treatment tool.

## 4. Discussion and Conclusion

Notably, flexible wearable sensors that are used in biomedicine are different from the traditional sensors that are used in signal processing. Owing to the particularity of the working environment of wearable sensors and the miniaturisation of devices, wearable sensors often use integrated back-end circuits in order to analyse signals. Microcontrollers are usually used for signal acquisition and transmission, combined with the C language and Python, in order to develop PC-based applications for data visualisation. The real-time display on the terminal is realised through the interaction with the application program through serial communication. In wireless sensing devices, coil coupling, or optical communication is often used in order to realise data transmission. A real-time display on the mobile terminal can be realised using Bluetooth. The development of vector machine-supervised learning algorithms, deep learning algorithms, and applications is also extremely important in the medical Internet of Things.

Flexible wearable sensors for biomedicine can realise force detection in high-strain soft tissues, including human skin, the detection of temperature and humidity, and the detection of physiological information in complex environments in vivo. Different flexible wearable sensors are prepared using different materials, structures, sensing mechanisms, and communication methods, which can realise the stable and accurate detection of human pressure, strain, multi-angle force, vibration, temperature, humidity, cell tissue, flow, and microorganisms. Therefore, we believe that the application of flexible wearable devices in biomedical applications is feasible.

In this review, we have presented an updated overview of wearable devices that are used in biomedical applications. Starting with materials, we have presented cutting-edge preparation techniques and have analysed the important properties of these materials. By relying on advanced material preparation techniques, we have combined modern sensing technologies and MEMS processes in order to design and fabricate excellent sensor devices. We have discussed the unique sensing modalities, the excellent performance, and some exciting applications of these sensor devices from the perspective of in vivo and in vitro sensing. To date, these wearable devices have exhibited remarkable performance in solving specific problems with a much greater efficiency. This emerging field encompasses considerable opportunities and challenges in terms of the sensing capability, orientation, and adaptability. Flexible wearable sensor devices are increasingly moving toward adaptive, self-feeding, and personalised directions, providing multimodal [144], multifunctional [145], and multidirectional [146] detection capabilities. This vision has been made more relevant by combining it with data-processing and wireless-transmission systems. However, the types of signals that sensors can perceive are still limited. For example, the effect of in vitro detection at the cellular and protein levels are poor. The active detection of sensor targeting in vivo [147,148] remains a problem that modern science urgently needs to overcome. There is considerable room for the optimisation of the compatibility between wearable devices and human tissue, from harmless to beneficial, from short-term to long-term monitoring, and the optimisation of the degradation time. In the future, the development of wearable devices will require the cooperation of various disciplines (materials science, biomedicine, chemistry, microelectronics, and communication technology) in order to break professional barriers. It is worth noting that the development of processing technology and testing technology is equally important in order to realise the leap from theory to reality. Flexible wearables are the result of a multidisciplinary intersection. With the rapid development of various disciplines, it is reasonable to predict that flexible wearables will definitely attract attention in biomedical applications.

## 5. Summary and Outlook

We have presented the latest research on various flexible wearable sensing devices for biomedical applications, focusing on in vivo and in vitro sensing. As mentioned earlier, the detection of human physiological signals has expanded with the development of new materials and advances in sensing technologies. Flexible wearable sensors have great development potential in the fields of assisted diagnosis and treatment, health monitoring, cell capture, medical prosthetics, human–computer interaction, drug delivery, and targeted therapy.

Future work and product promotion face many obstacles. Whether in vitro or in vivo sensing, the known materials still cannot fully meet the needs of detection, which is closely related to the weakness of human signals. Discovering more emerging materials, performing different degrees of synthesis, doping on known materials, and exploring more suitable materials will be beneficial. For example, combining different groups with MXene and MOF materials, or crosslinking and modifying hydrogels, can achieve better performance. The development of various adaptive and self-powered materials has significantly boosted the development of implantable sensors. In addition, the shortcomings of sensor communication technology restrict the detection of human physiological signals, which is especially obvious in in vivo sensing. There are three reasons for this finding, as follows: First, human tissues have certain obstacles to signal transmission. Second, the complex environment of the body is not conducive for the operation of the sensor. Third, the rejection reaction of the human body itself, as well as the material properties of the sensor, can easily cause discomfort to the human body. Additionally, the development of processing and testing technologies is equally important for the development of new materials, new structures, and more sophisticated wearable devices. The rise in 3D printing technology has led to the refinement of device products and a higher yield. Simultaneously, the development of medicines is extremely important for drug delivery and targeted therapy.

Notably, the detection of human physiological information using wearable devices has great application potential in the medical Internet of Things. Through wearable device protocol communication technology, the IoT gateway is constructed, and the IoT server realises the real-time communication between the users and the medical staff. Medical staff can detect physiological information based on the wearable devices, set detection parameters, and synchronously process the user’s physical condition. The data visualisation of the wearable sensor on the mobile terminal through Bluetooth, which has been realised at this stage, has explored the architecture of medical IoT. Whether the wearable sensor comprehensively detects the physiological signal is directly related to whether the medical staff can make an accurate judgment. The architecture of the protocol communication technology, the IoT gateway, and the server must be jointly developed by researchers in many industries.

The application of flexible wearable devices in biomedicine requires joint development in various disciplines. It is believed that with the efforts of various disciplines, the application of flexible wearable devices in biomedicine will become more comprehensive. The transition of these applications from the laboratory to clinical applications and to actual life is something that is worthy of further study.

## Figures and Tables

**Figure 1 micromachines-13-01395-f001:**
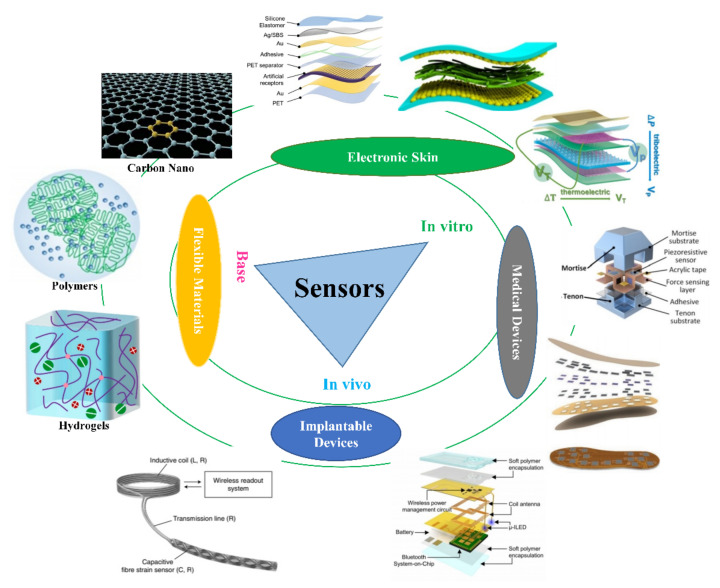
Overview of research in flexible wearable sensor devices for biomedical applications.

**Figure 2 micromachines-13-01395-f002:**
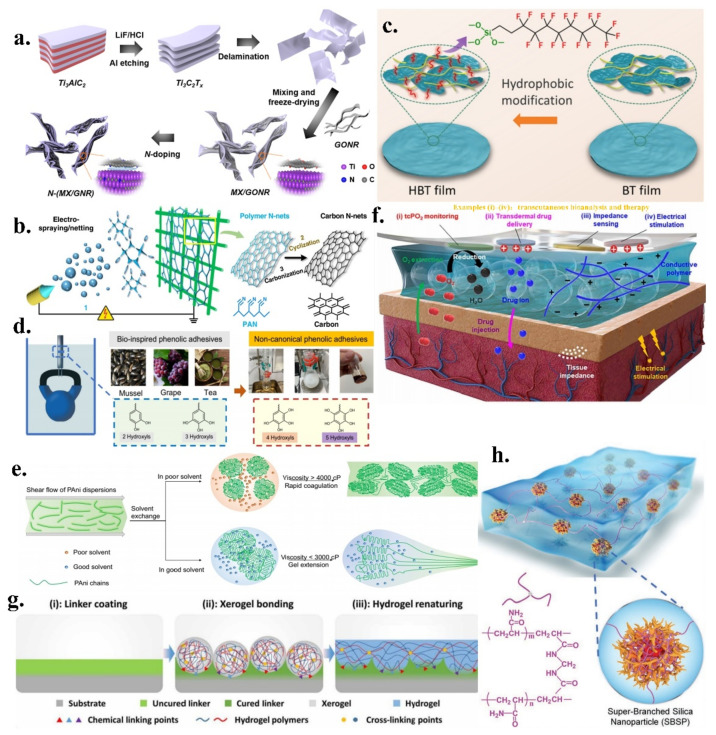
(**a**) New dopant-induced hybridisation method for 1D/2D materials; reproduced with permission from the American Chemical Society (2021) [43]; (**b**) Self-assembled 2D carbon nanostructure network; reproduced with permission from Springer Nature (2020) [44]; (**c**) Ti3C2Tx-type MXene film that is resistant to oxidation; reproduced with permission from the American Chemical Society (2022) [45]; (**d**) Non-standard phenolic polymer; reproduced with permission from Springer Nature (2022) [46]; (**e**) Ultra-fine polyaniline fibre; reproduced with permission from Springer Nature (2022) [47]; (**f**) Ultra-soft, highly permeable, low-impedance ultrathin hydrogel; reproduced from Ref. [48]; (**g**) Method for adhering hydrogels to various solid interfaces; reproduced from Ref. [60]; (**h**) Highly solvated, hyperbranched nanoparticle-reinforced polymer hydrogel; reproduced with permission from Wiley (2022) [49].

**Figure 4 micromachines-13-01395-f004:**
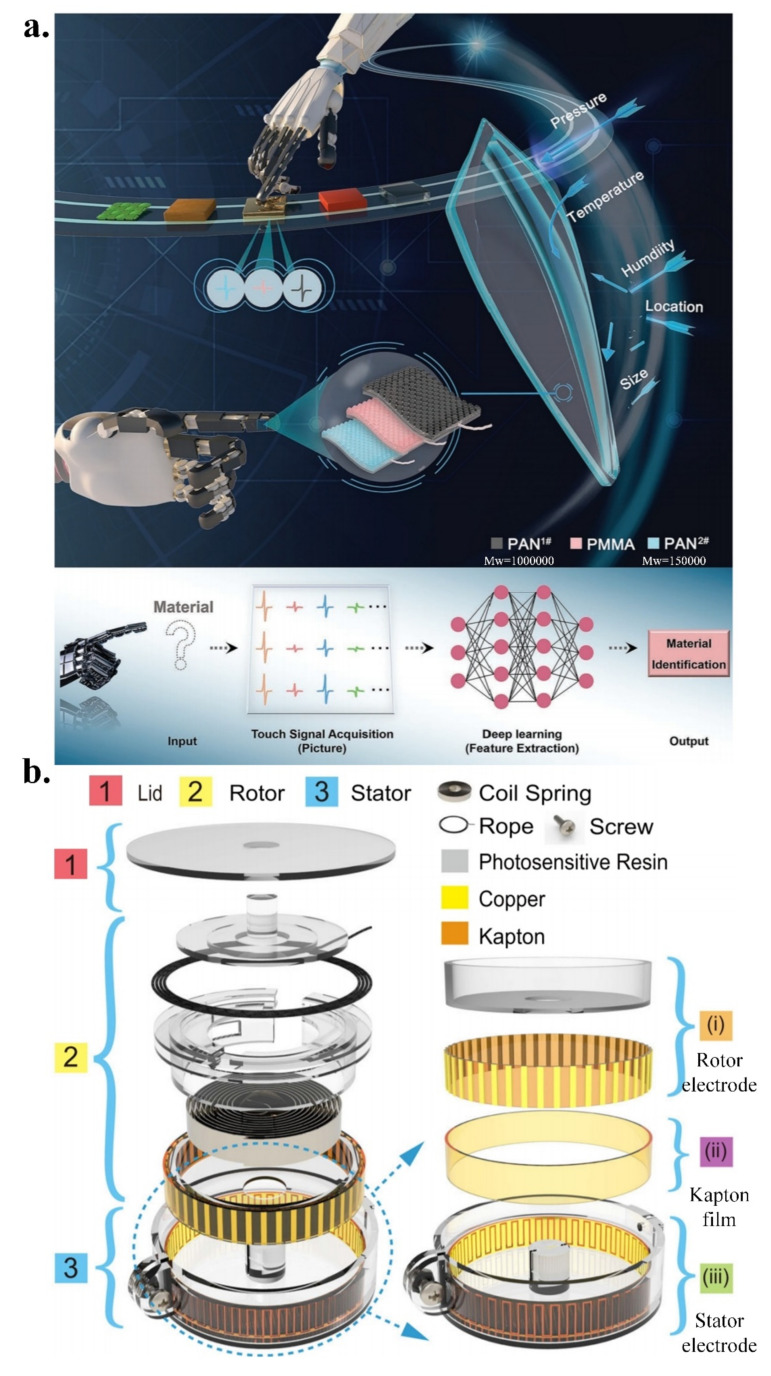
(**a**) Intelligent tactile sensing system combining TENGs and deep learning technology; reproduced with permission from Wiley (2022) [105]; (**b**) Badge-reel-like, stretchable, wearable, self-driving sensor, and its system; reproduced with permission from Springer Nature (2021) [106].

**Table 1 micromachines-13-01395-t001:** Preparation methods, advantageous properties, modes of action, and some unique applications of synthetic materials.

Selection of Materials	Carbon Nanomaterials			Polymeric Materials		Hydrogel Materials	
	Graphene Nanoribbons in 1D, Ti3C2Tx MXene in 2D	Carbon Nanotubes in 1D, Graphene in 2D	1H,1H,1H, 2H-Pefluorodecyltrimethoxysilane, Ti3C2Tx Mxene	Styrene Monomer, Styrene Radical, Hydroxyl Group	Polyaniline, Dimethyl Formamide	PAAm Hydrogel, 3,4-Ethylenedioxythiophene, and Polystyrene Sulfonate	SiO_2_ Nanoparticles and PAAm Hydrogel
Synthesis and preparation methods	Doping-induced hybridisation, nitrogen doping	Controlling the dynamic injection of charged droplets	Superficial fluorine functionalisation, grafting modification	Solution polymerisation	Solvent-exchange strategy, wet-spinning technology	Dissolution diffusion	Adjustment of network structure and cross-linking mechanisms
Advantageous performance	High cycle stability, low hysteresis, durability	Nanoscale structure, nanoscale conductivity, lateral infinity	Surface hydrophobicity, resistant to oxidation	Powerful underwater absorbency, durability	High tensile ratio, high energy, excellent mechanical properties	Porous structure, ultra-thin thickness, high permeability, low impedance	High elasticity, no temporary entanglement, high sensitivity
Mode of action	Through the synthesis, doping, and modification of materials, the advantageous properties of multiple materials are combined, and new properties are generated through reactions, resulting in excellent composite materials, sensing media, and interactive interfaces.						
Applications	Improved low-dimensional material properties for health detection systems	Pseudo-3D macroforms and core fillers for high performance nanocomposites	Perception in a liquid environment, as a sensing medium layer	Bonding of devices in liquid environments	Weaving, energy harvesting, and charge storage	Acts as a liquid electrolyte, forming a conformable and low impedance interface	Used in repetitive motion with minimal energy dissipation
Main weaknesses	Inherently fragile	Easily tied a knot	Inherently fragile	Steric hindrance between hydroxyl groups	Easy to break	Poor mechanical properties and easy to tear	Not close contact
References	[43]	[44]	[45]	[46]	[47]	[48]	[49]

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
