# Peer review of "Review of Flexible Wearable Sensor Devices for Biomedical Application"

_micromachines, 2022, doi:10.3390/mi13091395_

Round 1
Reviewer 1 Report
The authors present an insightful review on Review of Flexible Wearable Sensor Devices for Biomedical Application. The topic is interesting and useful. It contains plenty of the recent progress in flexible sensors. I have some suggestions, though:
1. In the introduction, the Authors should comment on the market shares and presently used flexible wearable sensor devices for biomedical applications
2. Table1: Among various materials, authors have selected a few of them. Please explain your decision in considering only these materials.
3. Authors should add one more widely used thin film form based materials for flexible devices in a different section with proper explanations.
4. At the last of every figure caption references, authors should write reproduce with the permission from ACS/Wiley/Elsevier/..other...
5. The recent works based on the sensors for flexible devices should be cited in the appropriate place of the articles https://doi.org/10.3390/s22124460.
6. The disadvantages and challenges of the all methods explained here should be explained in a different section with future perspectives- "Challenges and future perspectives".
7. In overall content, I find that summary of articles. Authors need to discuss it in their understanding. Also, a connection between the paragraph is needed. Without studying the reference in detail, it is difficult to understand in this section exactly what the authors mean.
Reviewer 2 Report
This manuscript reports an overview of emerging flexible wearable sensors for biomedical applications. This manuscript includes the selection and fabrication of flexible materials and the working mechanisms of flexible wearable devices. However, this manuscript must be improved considering the following comments:
1.-English grammar and style of all the manuscript sections must be revised.
2.-Introduction should consider the advantages and limitations of the flexible wearable sensors for biomedical applications.
3.-Resolution and quality of Figures 1, 3, 4, 5, and 6 must be enhanced.
4.-Table 1 should include the main weaknesses of the different synthetic materials.
5.-The authors should add more detailed information on the advantages and limitations of the material types reported in the second section.
6.-The working mechanisms of the different flexible wearable devices reported in the third section should consider more detailed information.
7.-All the figures must include the copyright permissions of the papers.
8.-This manuscript should consider a section on signal processing of flexible wearable devices for biomedical applications. In addition, this manuscript should add a section on the challenges and perspectives of flexible wearable devices for biomedical applications.
9.-The authors must include discussions on the reliability of flexible wearable devices for biomedical applications. In addition, this manuscript should consider a sub-section of recent self-powered wearable devices using nanogenerators.
10.-This review should discuss the advantages and challenges of the internet of medical things considering the flexible wearable devices.
Round 2
Reviewer 2 Report
This manuscript was improved considering the reviewer's comments. This revised manuscript can be accepted for publication in Micromachines.